# DC-AC Isolated Power Converter Array. Focus on Differential Mode Conducted EMI

**Jean-Christophe CREBIER [1,2,\*], Thanh-Hai PHUNG [1], Van-Sang NGUYEN [1], Theo LAMORELLE [2], Andre ANDRETA [2], Lyubomir KÉACHEV [1] and Yves LEMBEYE [2,\*]**

[1] Université Grenoble Alpes, CNRS, Grenoble INP, CMP, F-38000 Grenoble, France
[2] Université Grenoble Alpes, CNRS, Grenoble INP, G2Elab, F-38000 Grenoble, France
\* Correspondence: jean-christophe.crebier@g2elab.grenoble-inp.fr (J.-C.C.);
  yves.lembeye@g2elab.grenoble-inp.fr (Y.L.); Tel.: +334-7657-4617 (J.-C.C.)

**Abstract:** This paper introduces the implementation of a DC–AC step up isolated converter from associations of bidirectional Conversion Standard Cells (CSCs). The designed multi-cell converter is an array of standardized converter cells. It is described and then compared to a reference converter with respect to differential mode conducted electro-magnetic interference (EMI). The paper outlines the motivation for developing a generic multi-cell approach before underlining the benefits from the point of view of conducted EMI when implementing power converter arrays (PCAs). In particular, it is shown that in PCAs, the differential mode (DM) EMI filter can advantageously utilize distributed CSCs, making it possible to use very low value AC inductors to filter the AC current ripple. Experimental results are provided to validate the analysis carried out in the paper.

**Keywords:** multi-cell converter; power converter array; bidirectional converter; isolated converters; DC to AC conversion; conducted EMI; differential mode currents

## 1. Introduction

Isolated DC-AC step up bidirectional converters are presently of great interest for connecting storage batteries to the grid for many applications, such as electric mobility and remote applications, as well as renewable energy storage [1–3]. These applications generally require an inverter/rectifier stage, as well as a DC-DC step-down/up stage to adapt the voltage rating of the battery. Isolation is commonly necessary to ensure user safety when the battery must be a removable unit or when its connection to the grid can be done manually. When the batteries connected to the grid exchange energy in both directions, a bidirectional energy flow is required. Since on the battery side the voltage remains fairly stable throughout the charge/discharge cycle, the dual active bridge (DAB) topology is well adapted [4]. For the inverter/rectifier stage, the full bridge voltage inverter is the best choice due to its high total harmonic distortion (THD) rejection capabilities [5], ease of use and well-known operating principle [6]. Optimizing the design of such a topology relies on a good tradeoff between switching frequency and passive storage and filtering components [7]. The DC bus between the two conversion stages is generally set to approximately 400 V and frequently receives, for single-phase application, a capacitor tank to filter the 100–120 Hz pulsed power. With a 400 V DC bus, the DC-AC full bridge imposes important electro-magnetic compatibility (EMC) filtering constraints for both differential mode and common mode currents. Moreover, this relatively high voltage level imposes significant constraints on the DAB transformer optimization.

In order to reduce these constraints, another way to implement this cascade conversion is to divide the converter into several low voltage conversion stages called conversion standard cells (CSC). The CSCs are then associated to produce the complete converter. The resulting power converter

array (PCA) is also called a multi-cell converter (MCC). The basic topology of the CSC is presented in Figure 1a. It presents a two-stage topology cascading a voltage inverter made with an H-bridge with a conventional DAB converter including an high frequency (HF) transformer. The CSCs can be connected in series on the AC side and in parallel on the DC side, as shown in Figure 1b. The series connection of the CSCs' input terminals permits the PCA to be adapted to the grid voltage while using low voltage CSCs. On the battery side, the output terminals of the DC-DC conversion stages are in parallel in order to produce a converter with a high transformation ratio. This configuration is also called input parallel/output series (IPOS), considering the DC side as the input of the converter. It is important to note that the AC-side inductor is always split on both AC terminals. In this way, the filter remains symmetrical, which avoids production of additional disturbances due to common mode coupling. This must be implemented in all cases as illustrated in Figure 1 in order to maintain, as much as possible, symmetrical EMI propagation paths.

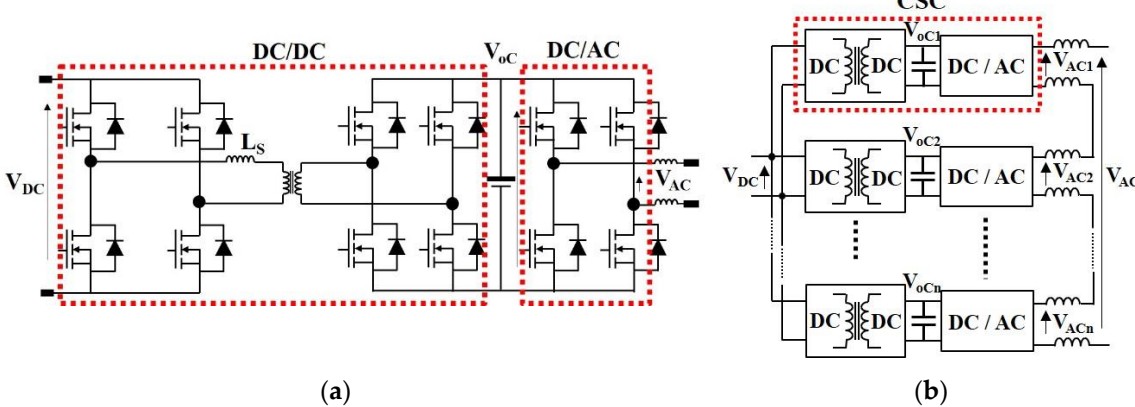

(**a**) (**b**)

**Figure 1.** (**a**) Topology of a single Conversion Standard Cell (CSC); (**b**) IPOS (input parallel/output series) configuration of the power converter array (PCA).

The design and optimization of such a power converter array depends, among other parameters, on the nominal voltage of the CSC DC bus and the battery. Since battery voltage levels are very application dependent, the nominal voltage of CSCs can be adjusted accordingly. If necessary, it is possible to adjust the transformer transformation ratio, thus offering the possibility to further increase or reduce the voltage ratio through the DAB conversion stage. The procedure for optimizing a multi-cell converter is described in detail in [8]. However, there are some advantages in maintaining a unity transformation ratio for the DAB HF transformer. First, transformers are easier to supply. Second, all CSC components have the same voltage ratings, on the primary and secondary sides. This enables the implementation of the same three H-bridges comprising each CSC, which concentrates design and optimization efforts on a single H-bridge layout, component selection and the supply chain. These considerations are important because many CSCs are going to be implemented, which scales up component quantities and scales down component supply costs. Although outside the scope of this paper, cost issues are a driving criterion in justifying the design and implementation of a generic and standardized conversion cell that can be optimized and reused in many different PCAs [9].

In terms of optimization, many aspects are of importance. Some are already addressed in [8,10] and some remain to be investigated. While conducted electro-magnetic interference (EMI) benefits have been already investigated in many multi-cell converter topologies [11], conducted EMI in the specific case of PCA remains to be addressed. In particular, the opportunity to spread the filtering needs over the CSCs and the resulting possible benefits are a very specific topic related mainly to PCA topologies [12,13]. The present paper focuses on the conducted EMI analysis of such a converter when several CSCs are implemented according to Figure 1. Since several stages are cascaded, paralleled and serried, the EMI signature becomes a combination of several contributions. The paper investigates,

from both theoretical and experimental points of view, the benefits of such a power converter array compared to the reference topology, which is the H-bridge plus a DAB converter. The objective is mainly to identify differential mode (DM) AC side filtering stage optimizations. In the first section, the operation principle of the PCA is outlined and the main design and optimization issues are briefly discussed. The second section focuses on conducted EMI analysis and, more specifically, DM current disturbances. The third section is dedicated to practical implementation and characterization, including a comparison with the theoretical estimates. In all sections, the PCA is compared to the reference topology. The last section provides analysis and comments on the obtained results.

## 2. DC-AC Power Converter Array

DC-AC converters from conversion cell associations have several interesting features. Among these, the distribution of voltage and current over several subsystems is a noteworthy advantage because it makes it possible to "distribute" losses but also EMI disturbances sources. The distribution of losses makes it possible to avoid the use of heat sink as illustrated in [9] and [12]. Regarding EMI signatures, the combination of lower nominal voltages and/or currents with interleaved control produces switching events with lower magnitudes that are distributed over the switching period, leading to higher apparent switching frequency, equals to the converter cell switching frequency multiplied by the number of CSCs implemented. Compared to a full bridge converter, the magnitudes of the voltage and/or current ripples are divided by the number of CSCs. These are well known benefits of multi-cell interleaved converters as illustrated in [14].

In [13], the authors analyzed the opportunity of distributing the EMI filters in CSCs to make them more generic from the design and implementation points of view. This section analyzes the opportunities to implement a PCA, minimizing its filtering needs. Indeed, lower nominal voltage and current ratings bring the opportunity to produce an integrated power electronics module (IPEM), including most of the components of each CSC [15]. Hybrid and monolithic integrations are used to size down and to optimize conversion subsystems. In addition, integration provides the opportunity to significantly increase the switching frequency of each CSC since active components with lower ratings can be used, and are usually able to switch at much higher frequencies. Increasing the switching frequency is known as a good first-order option to reduce filtering needs. The combination of a high switching frequency with a high number of CSCs to be implemented opens up the opportunity to also integrate the DM filter in the CSC.

### 2.1. AC Side DM Mode Filter Design Considerations

In voltage inverters, the EMI filter on the AC side starts with two inductors, one on each phase. The inductors are filtering the HF voltage patterns produced by the H-bridge. Depending on the switching frequency, a large value of inductor is required to reduce the current ripple as a portion of the nominal AC current. Usually, it remains difficult to increase the switching frequency in grid-connected voltage inverters due to the hard switching operation of the switches that produces losses proportional to it. As a result, the DM filter generally includes several passive components to achieve a high-order filter, making it possible to remain below regulation limits.

In DC-AC power converter arrays, when converter cells are associated in series on the AC side, as presented in Figure 1b, the AC inductors, when located on each CSC, are summed. Since the DC bus voltage at the CSC level is lowered by the number of CSCs to be interleaved, the inductor value is reduced accordingly. As a result, the large AC inductor of the conventional H-bridge can be replaced by many small inductors integrated into each CSC of the PCA. Figure 2 illustrates a PCA including a distributed AC side DM filter with pairs of inductors on each CSC. Table 1 provides a set of AC side inductor values for several PCAs with various numbers of levels, maintaining the DC bus at a total voltage equivalent to 400 V and keeping the magnitude of the ripple current at 10% of the nominal AC current magnitude.

Two positive effects are clearly visible in Table 1. First, by increasing the number of CSCs, i.e., the number of levels, the total equivalent AC inductor value (with interleaving control) is significantly reduced by a factor equals to (number of CSC)$^2$. For example, for 20 CSCs, the total AC inductor falls from 10 mH to 25 µH. Second, by distributing the 25 µH over the 20 CSCs and considering that two inductors are placed at the input terminals of each CSC, the value for one inductor falls to 625 nH. Such a low value for the AC inductor is easier to supply (parts are available off the shelf) but also to integrate within the CSC. Other benefits will be underlined later in the paper.

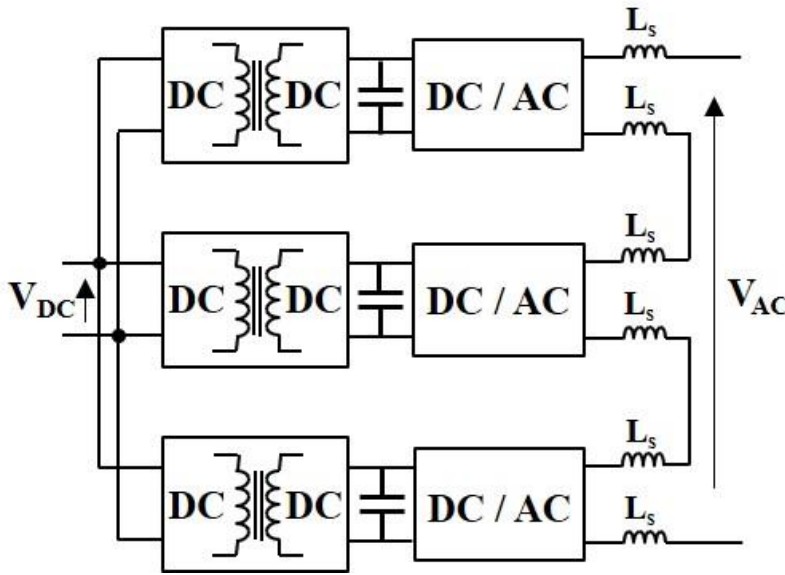

**Figure 2.** Topology of a PCA with three CSCs including their distributed AC inductors.

### 2.2. Design Issues

Interleaving techniques are well-known control strategies to reduce current ripples and to produce higher apparent switching frequencies, proportional to the number of interleaved conversion cells [16]. Table 1 shows the impact of the interleaved control on the apparent switching frequency and the value of the needed inductor to keep the ripple current below 10% of the nominal current. This 10% is an arbitrary level in order to limit current ripple in the inductors and to maintain losses due to skin effects at reasonable levels. This is done with the sole purpose of illustrating the benefits of the PCA over the H-bridge converter and is not intended to provide a final optimization with respect to regulations, efficiency, cost or power density. The nominal input current is considered equal to 5 A max. Therefore, the maximum ripple current, $\Delta I_{AC}$, must be kept lower or equal to 0.5 A in this case.

**Table 1.** Values of AC inductors for PCA with different numbers of CSCs. CSC individual switching frequency equals 40 kHz. Equivalent DC bus equals 400 V. Basic interleaving control whenever applies.

| Number of CSCs | CSC DC Bus Voltage Ratings (V) | AC Side Inductor Value Per CSC (µH) | Apparent Switching Frequency (kHz) | Total AC Inductor with Interleaving (mH) |
|---|---|---|---|---|
| 1 | 400 | 2 × 5000 | 40 | 10 |
| 3 | 133 | 2 × 185 | 120 | 1.1 |
| 6 | 67 | 2 × 23 | 240 | 0.28 |
| 20 | 20 | 2 × 0.625 | 800 | 0.025 |

CSC DC bus voltage, $V_{DC}$, is derived by dividing the reference DC bus voltage (400 V in this case) by the number of CSCs. The value for the apparent switching frequency, $F_{SA}$, is derived by multiplying the reference switching frequency (40 kHz in this case) by the number of CSCs. The AC side inductor

value, $L_{AC}$, per CSC is derived using Equation (1), where the factor 0.5 accounts for the worst-case duty cycle with respect to ripple current magnitude. The total AC inductor considering interleaved control is obtained by multiplying the AC side inductor value per CSC by the number of CSCs. As an example, with 20 CSCs, $20 \times 2 \times 0.625$ nH makes a total inductor on the AC side equal to 25 µH. In this example, the factor of 2 comes from the fact that two inductors are implemented per CSC, one on each input terminal.

$$L_{AC} = V_{DC} \times \frac{0.5}{\Delta I_{AC} \times F_{SA}} \tag{1}$$

In the following section, the AC side conducted DM mode currents generated in a PCA are forecast and compared to the one produced by the reference converter and are analyzed in more details.

## 3. Conducted Differential Mode Current Analysis

For commercialization purposes, conducted differential mode EMI levels must be maintained within regulation limits, among other constraints. On the AC side, the produced harmonics must remain below the limits specified in EN55022 regulation [17]. The conducted EMI sources produced by any voltage source inverter can be theoretically and accurately estimated for low order HF harmonics as illustrated in [18,19]. For higher-order HF harmonics, the estimates provide tendencies. The cited modeling technique can also be applied to PCA topologies as will be shown below. Although the estimates produced are based on first-order modeling techniques, they can be used to compare converters with respect to their conducted EMI signature. Propagation paths are also important to represent at best not only the EMI sources but also the HF currents that will be produced by any converter. In the upper frequency range, many parasitic effects are very challenging to represent and to model, making it even more difficult to derive accurate predictions. The propagation paths to be considered include all components and parasitics between the EMI sources and the line impedance stabilization network (LISN) used to perform normative conducted EMI testing [17]. In particular, it is important that the filter stage is carefully represented as it greatly impacts on EMI levels measured at the LISN testing points [17]. The next section deals with the AC inductors used to filter the AC side current. It provides a methodology for estimating DM disturbance levels at LISN terminals.

### 3.1. About the Impact of Distributed AC Side Inductor in CSCs

Distributing the AC inductor over the CSCs is a good option for producing generic conversion blocs, ready to be combined. It has also been shown that thanks to multi-levels and interleaving, the total equivalent AC inductor can be significantly reduced (see Table 1). Although more components may result in additional costs and reliability issues, this section highlights the potential benefits of distributing the total AC side inductor on each CSC. Indeed, the distribution of the AC inductor over numerous CSCs reduces the value of the inductor to be integrated in each CSC as shown in Table 1. In addition, a smaller inductor value provides an interesting advantage in the upper frequency range. Since inductors are smaller in value, their size, volume and surface area are also smaller. As a result, they have lower parasitic elements which, in turns, repels their resonant frequency at higher frequencies. This has been investigated with four inductors available in the lab or designed for the purpose of the project. Figure 3 provides a picture of the four components. One is a XEL5030–601ME device from Coilcraft. The three others have been designed and manufactured by a subcontractor for AC current in the range of 5 to 7 $A_{RMS}$ for switching frequencies in the range of tens of kHz. Table 2 provides, for the four inductors, the values of the R-L-C components of their equivalent parallel circuit, taking into consideration that, in the upper frequency range, the interwinding capacitance takes over the inductance effect. All these data are from experimental characterization carried out with an impedance analyzer. It is interesting to highlight how much the resonant frequency of the inductor is influenced by its value. This is not, of course, a discovery, but simply a demonstration of the benefits a small inductor value may bring in the upper frequency range. Table 2 also displays the value of the

series resistance of each component, which remains a critical parameter as far as conduction losses are considered.

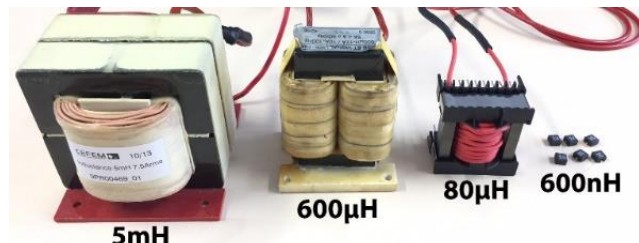

**Figure 3.** Picture of the AC inductors considered in this work.

**Table 2.** Component values of the equivalent parallel L-C-R circuits + series R for four inductor values. (nominal inductor current 5 A). Included, their resonant frequency and their volume. All data are from experimental characterization of the components using impedance analyzer and the automatic equivalent circuit function available in the menu of the equipment.

| Inductor Number | Inductor Value (H) | Parasitic Capacitance (pF) | Parallel Resistance (Ω) | DC Series Resistance (Ω) | Resonant Frequency (MHz) | Component Volume (cm³) |
|---|---|---|---|---|---|---|
| 1 | 5 m | 93.4 | 293.9 k | 0.165 | 0.236 | 810 |
| 2 | 600 μ | 26.8 | 84.6 k | 0.061 | 1.3 | 240 |
| 3 | 80 μ | 8.9 | 8.1 k | 0.017 | 5.5 | 67 |
| 4 | 600 n | <2.9 | 0.4 k | 0.003 | >120 | 0.22 |

Based on this study, the opportunity to spread the AC inductor over the CSCs is interesting for the upper frequency range. Indeed, using smaller inductor values, the AC inductor resonant frequency is shifted to higher frequencies, adding compliance with regulations in the upper part of the spectrum. By combining Tables 1 and 2, one can derive easily the equivalent characteristics of an AC inductor made with many small inductors. In Table 1, for 20 CSCs, a 25 μH total AC inductor is required to filter the input current ripple. Implemented with a 600 nH inductor on each input terminal of each CSC, 40 inductors are needed to reach 24 μH. Table 3 provides a comparison between case 1 with a single H-bridge using 2 AC input inductors of 5mH, and case 2 using 2 × 20 600 nH small inductors with a 20 CSC PCA to obtain the same input current ripple, equal to a maximum 10% of the nominal current e.g., 0.5 A.

**Table 3.** Comparison of the input current AC inductor value, size and main characteristics when implemented with two components in a single phase full bridge in case 1 and when implemented with two components in each of the 20 CSCs operated under interleaved control in case 2. Computation conditions identical to that in Table 1.

| Case Number | Expected Inductor Value (μH) | Number of Components | Resonant Frequency (MHz) | Equivalent DC Series Resistance (Ω) | Equivalent Component Volume (cm³) |
|---|---|---|---|---|---|
| 1 | 5000 | 2 | 0.25 | 0.331 | 1620 |
| 2 | 0.6 | 40 | 120 | 0.12 | 8.8 |

Of note, Table 3 shows the effect of the distribution of the AC inductor of the input filter over the CSCs in order to improve filtering characteristics. Thanks to the multi-level topology, interleaving control technique, and component distribution over the CSCs, the volume of the total AC side inductor is over one hundred times smaller and has a total equivalent DC series resistance more than two times smaller. In addition, the equivalent resonant frequency of the AC inductor is two decades higher, which considerably increases the filtering capabilities of the AC inductor in the upper frequency range.

These AC inductors, combined with a global capacitor filter, can provide a highly efficient conducted DM disturbance filter for regulation compliance. This point is investigated in the following section using frequency domain representations of the DM disturbances produced by the reference topology and the proposed PCA topology.

### 3.2. AC Side Differential Mode Signature Modeling

In [20,21], a simple and effective modeling approach has been described to estimate conducted differential mode EMI sources from a boost-derived converter, such as the H-bridge converter. The approach, based on the Laplace transform, has been used to derive Equation (2) which represents the EMI source for a single-phase H-bridge operated under unipolar or bipolar control strategies. By combining this frequency representation of the HF disturbance source with the propagation path from the converter to the LISN, DM currents can be estimated. The propagation path considered in this work is represented in Figure 4, where $U_{DM}$ is the disturbance source, and R1 and R2 are the two LISN measurement impedances. The propagation path can then be expressed in complex form as an equivalent impedance divider from the source to the LISN measurement points. Converting the voltage applied to the LISN resistor into dBμV, the estimates can be compared to standard regulations:

$$U_{DM}(n) = [\sum_{m=1}^{k} \frac{V_{DC}}{i \times n \times \omega} \times (1 - e^{-i \times a(m) \times To \times n \times \omega}) \times e^{-i \times m \times To \times n \times \omega}] \times \frac{1}{T} \qquad (2)$$

$$a(m) = D \times (1 + sin(\omega \times \frac{m}{k}))/2 \qquad (3)$$

where $n$ is the multiple of the HF harmonic, $\omega$ the switching frequency pulsation, $To$ the switching period, $T$ the line period, $i$ the complex imaginary operator, $k$ the ratio between the switching period and the line period, $V_{DC}$ the H-bridge DC bus voltage level and $a(m)$ the H-bridge duty cycle over a line period in unipolar control strategy with $D$ the magnitude of the sine carrier (between 0 and 1).

Figure 4 represents the simplified equivalent circuit between the H-bridge and the LISN for the differential mode. The propagation paths include the DM filter made with inductors and a Cx capacitor. For the values of the LISN components, please refer to regulations EN55022 [17]. The grid is represented here by a pure resistor. Its impact is very limited, hidden by the LISN in the upper frequency range.

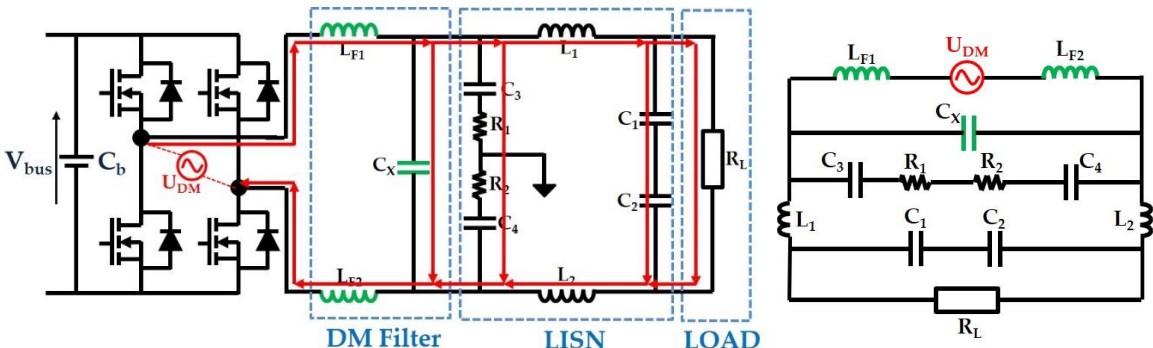

**Figure 4.** DM (Differential Mode) conducted disturbance propagation path equivalent circuit from the H-bridge to the LISN (Line Impedance Stabilization Network) measurement resistors $R_1$ and $R_2$.

Similarly, the disturbance source estimate equation and the propagation path equivalent circuit can be deduced for a PCA with a variable number of CSCs. Equation (4) gives the frequency representation of the disturbance sources in the case of a PCA, where *nbc* represents the number of CSCs. This equation is derived from Equation (2), summing up the contributions of *nbc* CSCs, considered to be interleaved.

Figure 5 shows the equivalent circuit between the equivalent disturbance sources and the LISN from which the complex representation of the voltage divider can be easily derived.

$$U_{DM}(n) = \sum_{g=1}^{nbc} \left[ \sum_{m=1}^{k} \left[ \frac{V_{DC}}{i \times n \times \omega \times nbc} \times (1 - e^{-i \times a(m) \times To \times n \times \omega}) \times e^{-i \times m \times To \times n \times \omega} \right] \times \frac{e^{-i \times g \times (\frac{T}{nbc}) \times n \times \omega}}{T} \right] \quad (4)$$

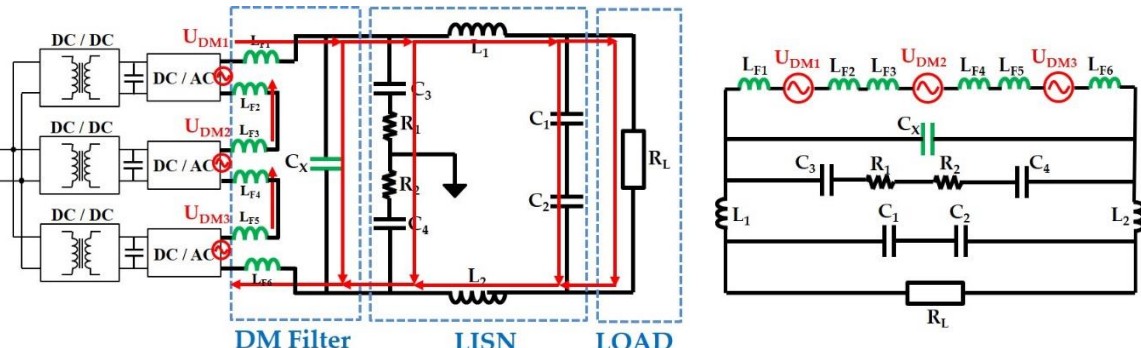

**Figure 5.** DM conducted disturbance propagation path equivalent circuit from the PCA to the LISN measurement resistors.

Figure 6 presents the estimates of the maximum envelop of HF harmonic distribution of the disturbance sources for both topologies. The red curve (with bullet) shows the envelope of HF harmonics produced by a conventional H-bridge with a DC bus voltage equal to 400 V. The magenta curve (+ cross) shows the result for a single CSC with a DC bus voltage at 20 V and the blue curve (with square) provides the HF harmonic envelop for 20 interleaved CSCs (with 20 V DC bus voltage), making visible the apparent shift of the switching frequency. These predictable slopes and respective attenuations validate, as is well-known and expected, the interest in interleaving H-bridges associated in series.

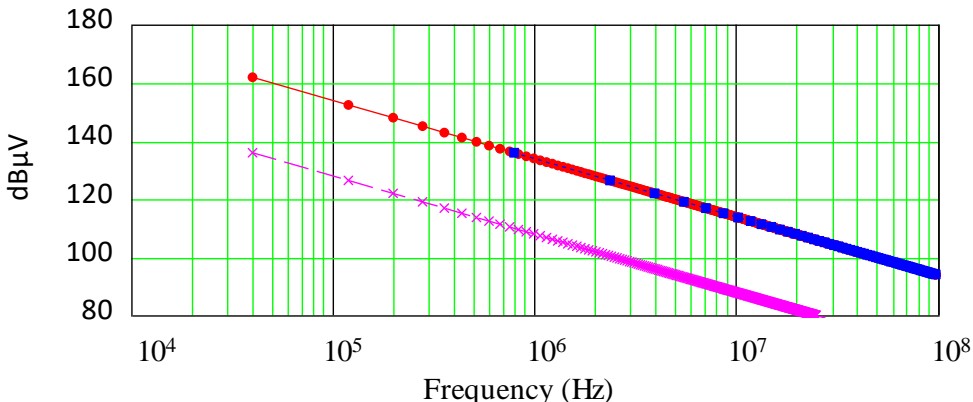

**Figure 6.** Estimated DM disturbance source harmonic levels in the frequency domain for several numbers of CSCs implemented. Red curve (with bullet) for the H-bridge with a DC bus voltage equal to 400 V, magenta curve (with cross) for a single CSC with a DC bus voltage at 20 V, and blue curve (with square) provides the HF harmonic envelop for 20 interleaved CSCs. Individual switching frequency equals 40 kHz.

Figure 7 shows the magnitude of the DM HF harmonics of the input current, as it could be measured on the LISN. In these computations, the propagation paths considered for the two cases are based on a specific component representation for the AC input inductor as presented in Table 2. In Figure 7, the magenta curves show the regulation limits. The red curve shows the magnitude of the

harmonics produced by a single-phase H-bridge converter including two 5 mH input inductors to comply with regulations. The blue curve shows the magnitude of the HF harmonics produced by the 20 CSCs operated under interleaved control, including $40 \times 600$ nH AC input inductors. In both cases C $\times$ capacitors are chosen, below or equal to 1 μF.

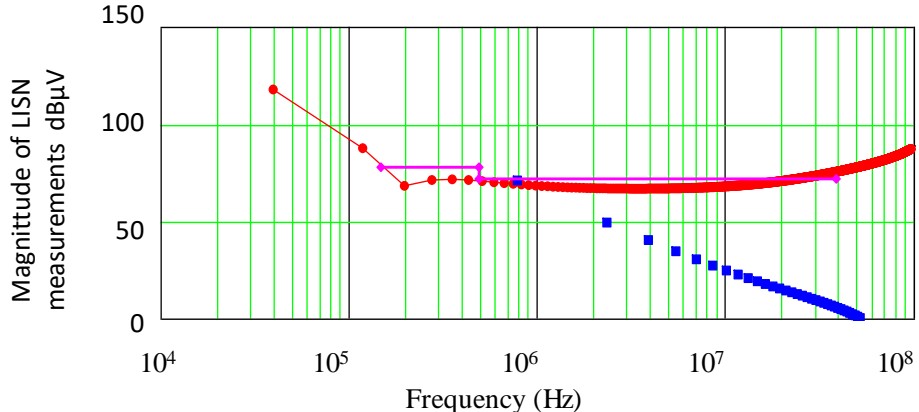

**Figure 7.** Estimated DM disturbance voltage level at LISN testing terminals in red for the single-phase H-bridge topology (with $2 \times 5$ mH AC inductors and 400 V DC bus voltage) and in blue for the PCA with 20 CSCs (with $2 \times 600$ nH AC inductors integrated in each CSC and 20 V DC bus voltage). Magenta curves provide the regulation limits.

In Figure 7, it is clear that both filter solutions almost comply with regulations. Nevertheless, the resonance frequency of the 5 mH AC inductors combined with the full bridge voltage source inverter is around 200 kHz. After this frequency, the filter attenuation is no longer active and the magnitude of the harmonics remains almost constant. A more complex and more accurate model may highlight harmonics over the regulation limits. With 20 CSCs and 40 small inductors, the first harmonic is just below the limit, around 800 kHz. Since the resonance frequency of the small inductors is much higher, the attenuation of the DM filter is effective up to much higher frequencies (above 100 MHz in this case), making this filter solution more reliable in the upper frequency range.

These promising theoretical results must now be verified in practice. This is presented in the next section.

## 4. Experiments

Conducted differential mode EMI must be kept below the regulation limits. On the AC side, the harmonics produced must remain below the limits specified in the EN55022 regulations. The conducted differential mode currents produced by the proposed PCA have been theoretically estimated and compared to those produced by the reference converter. The whole analysis was carried out at nominal voltage ($V_{AC}$ = 230 Vrms and $V_{DC}$ = 400 V) and current ($I_{AC} = 5/\sqrt{2}$) to provide the reader with data that are representative of real cases and more comparable to those in the literature. In this section, we will verify from an experimental point of view whether the expected benefits are real with a prototype implemented at a smaller scale. The specifications of the experimental set up are listed in Table 4.

**Table 4.** List of all testing conditions.

| Converter Type | Number of CSCs | Apparent Switching Frequency (kHz) | AC inductor Value (μH) | Cx Capacitor (μF) | AC Resistive Load (Ω) | Modulation Carrier Magnitude (D) | DC Bus Voltage (V) |
|---|---|---|---|---|---|---|---|
| Full bridge | 1 | 40 | $2 \times 600$ | 0.6 | 10 | 1 | 54 |
| PCA | 3 | 120 | $6 \times 30$ | 0.6 | 10 | 1 | $3 \times 18$ |

The values for the various parameters are derived in the same way as explained for Table 1. Figure 8 shows a picture of the test bench and one of the PCA prototypes. It can be seen that the converter is composed of three CSCs. More information about the prototype is available in [9]. In each CSC, a DAB converter is cascaded with an H-bridge having two AC inductors integrated, one on each of its AC terminals. Each CSC is a 20 V/5 A conversion block. DAB converters operate at 250 kHz with the same phase shift angle. H-bridges operate at 40 kHz switching frequency with the same PWM, 120° out of phase for each CSC. A 9 mF DC bus capacitor tank is decoupling the two conversion stages. It is represented in Figure 8 with 3 × 3 mF capacitors associated in parallel for each CSC.

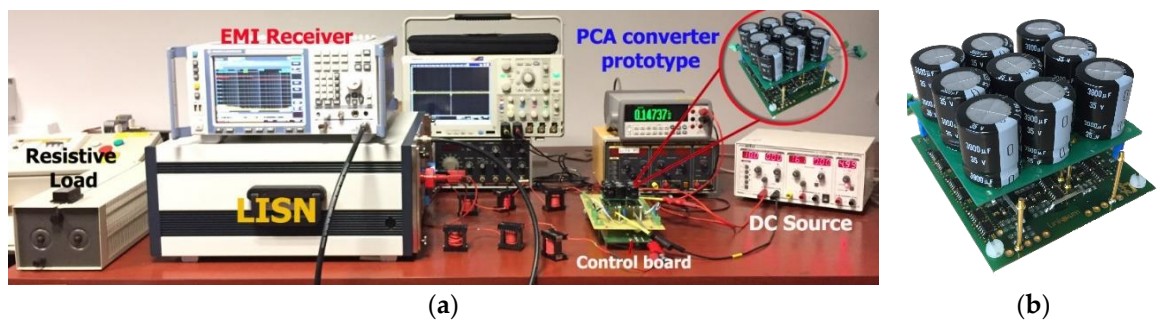

**Figure 8.** (**a**) Picture of the test bench. (**b**) Picture of the power converter array under test (the capacitor tanks of the three CSCs are clearly visible).

*4.1. AC Side Differential Mode Conducted EMI Signature of the Reference Converter*

The reference converter was first tested to provide a set of reference measurements. Below are shown three spectrum plots of the DM mode disturbance levels measured at the LISN measurement point. Figure 9 presents the spectrum produced by the H-bridge converter without any AC side inductor. In such a way, the measured current harmonics are directly proportional to the disturbance sources. Figure 10a presents the spectrum produced by the full bridge converter with the corresponding AC inductor, while Figure 10b presents the same measurement including the Cx capacitor to comply with regulations (see Table 4 for component and parameter details). The three plots give the experimental results in red, superposed with the theoretical spectrum estimates from the modeling approach introduced above. In black are added the regulation limits.

In Figure 9, the first harmonic, at the switching frequency (40 kHz) is attenuated due to the LISN impedance effects under 150 kHz. In the upper frequency range, the theoretical model is over-estimating the magnitude of the DM harmonics. This is mainly due to the first-order modeling that does not take into account the non-infinite slope during switching transitions, as well as the non-fully stable time location of the switching events. Apart from that, the tendencies are quite comparable with a −20 dB/dec slope over a large part of the frequency range.

In Figure 10, the measurements show well the cumulative impacts of the AC inductors and the Cx capacitors, both reducing significantly the magnitudes of the DM harmonics. An important error is visible on the first harmonic because the LISN attenuation is important under the lower border of the regulation frequency range (below 150 kHz). The final spectrum, Figure 10b, is almost in compliance with the regulations, except for the two first harmonics at the very beginning of the regulation bandwidth. This could be corrected by increasing the value of Cx capacitor. The resonant frequency of the AC inductors can also be well observed, around 1.3 MHz as introduced in Table 2. This parasitic effect stops filter attenuation and it can be seen in Figure 10b that, in the upper frequency range, the magnitudes of the harmonics almost reach the regulation limit. In this frequency range the theoretical estimates are lower than 40 dBµV.

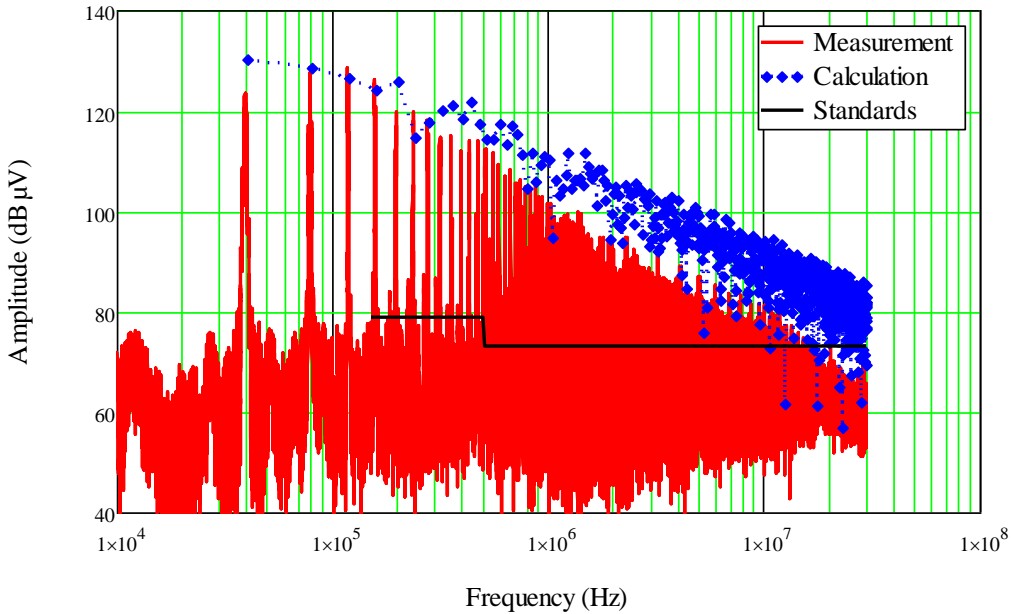

**Figure 9.** Red: measured DM disturbance harmonic levels at the LISN terminals for the H bridge converter without AC inductor (DC bus voltage equals 54 V). Blue: estimated levels. Black: regulation limits.

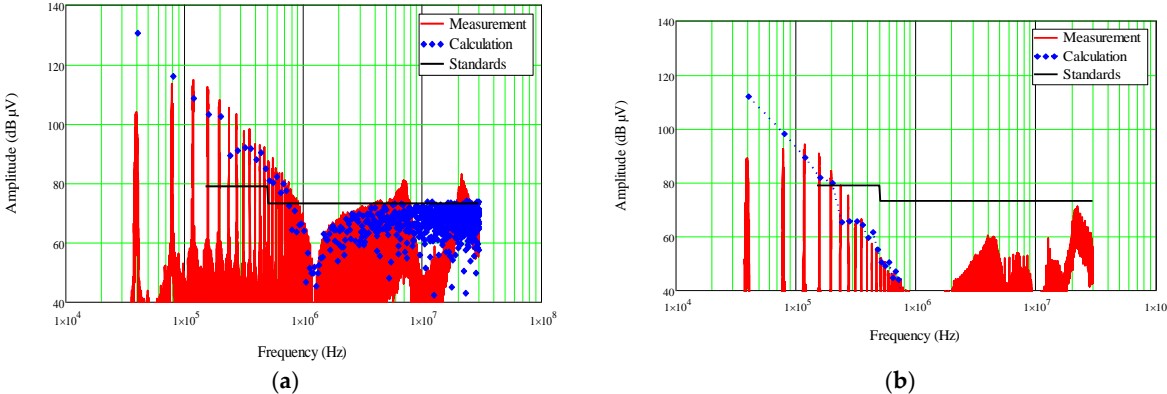

**Figure 10.** Red: measured DM disturbance levels at the LISN terminals with (**a**) the two 600 μH AC inductors and (**b**) the two 600 μH AC inductor and a 0.6 μF Cx capacitor. Blue: estimated levels. Black: regulation limits.

### 4.2. AC Side Differential Mode Conducted EMI Signature of the PCA

The PCA converter has been tested under equivalent conditions to study its DM conducted EMI signature. Below are presented the three spectrum plots of the DM mode disturbance levels estimated from a theoretical point of view and measured at the LISN measurement point. Figure 11 presents the spectrum produced by the PCA converter operated under interleaving control and without any AC side inductor. Figure 12a presents the spectrum produced by the PCA with the corresponding AC inductor, while Figure 12b presents the same measure including the Cx capacitor in order to comply with regulations.

In Figure 11, the impacts of multilevel and interleaving are visible. The first HF harmonic is at three times the switching frequency and the magnitude of the first harmonic is about the same as that at the same frequency in Figure 9. As can be seen in this Figure, the estimates are in satisfactory accordance with the experiments. Similar to what has been stated in the case of the H-bridge disturbance source

spectrum, the estimates are over the experimental results in the upper frequency range for the same reasons. Nonetheless, the tendencies are respected.

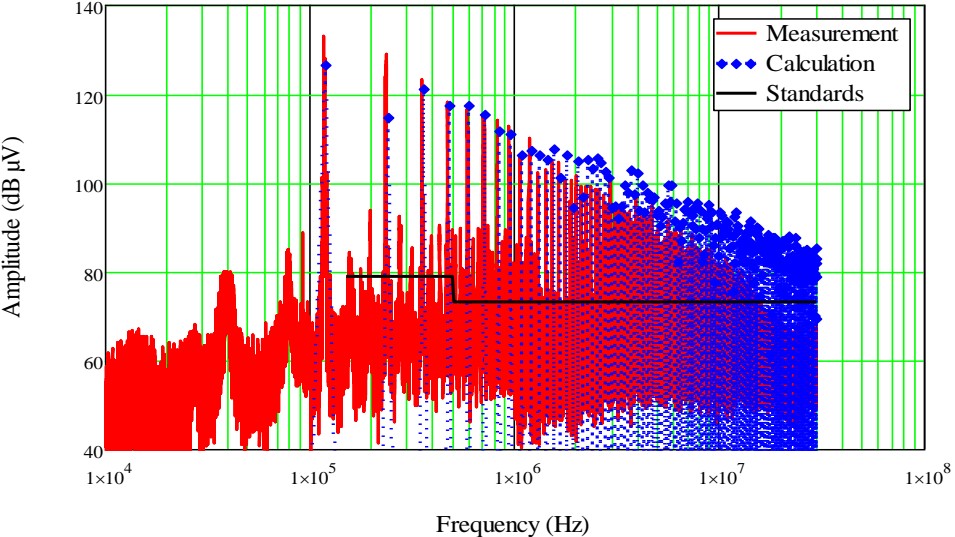

**Figure 11.** Red: measured DM disturbance level at LISN terminals for the PCA with three CSCs without AC inductor (DC bus voltage equals 18V + interleaved control). Blue: estimated levels. Black: regulation limits.

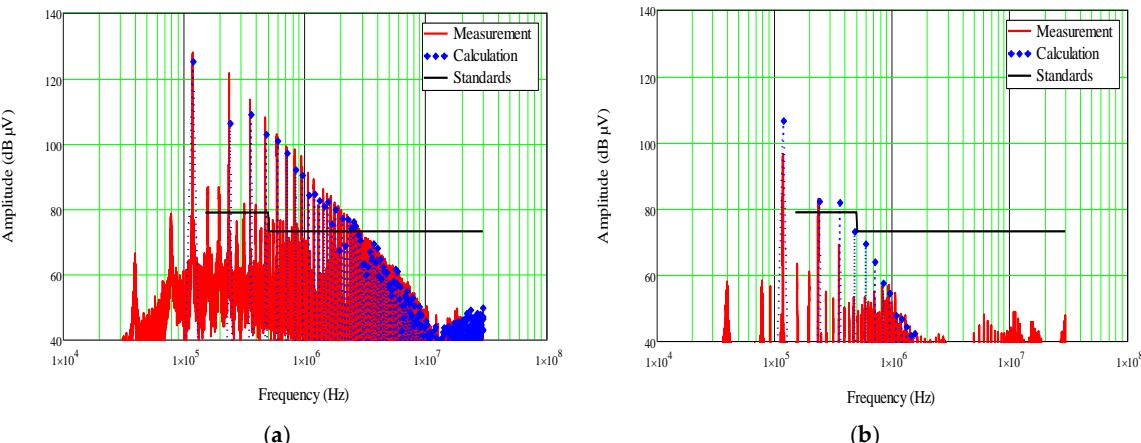

**Figure 12.** Red: measured DM disturbance level at LISN terminals with (**a**) the 3 × 30µH AC inductors, and (**b**) the 3 × 30 µH AC inductor and a 0.6 µF Cx capacitor. Blue: estimated DM harmonic levels. Black: regulation limits.

In Figure 12, compliance with regulations is also almost obtained. One can note that with multiple 30 µH inductors, the resonant frequency of the components is significantly higher in the frequency range, slightly above 10 Mhz. As a consequence, in the upper frequency range, the EMI levels remain below the regulation limits, which is positive compared to the previous case with the H-bridge converter and its two 600 µH AC inductors. This is especially interesting compared to what is presented in Figure 10b where it has been pointed out that in the upper frequency range harmonics were almost reaching the regulation limit. Globally, the estimates are in good accordance with practical results. They produce more significant errors in the upper frequency range because the model of the propagation path is not refined enough to take into account all possible parasitics.

*4.3. Analysis and Comments*

The results provided from the experiments are satisfactory. Even if the models do not match perfectly to the experiments, the trends are all confirmed. In addition, even if with our modeling and filter designs, the converters do not fully comply with regulations, the estimates have produced satisfactory tendencies to forecast the EMI spectrum and the components to add in order to comply with regulations in both cases, with the H-bridge and with the PCA. Compliance could be obtained simply by increasing the value of the Cx capacitor.

The main conclusions we can draw from this work are as follows.

First of all, as is well-known, multilevel interleaved converters offer great benefits in AC filter downscaling. This has been analyzed in theory and verified in practice. Second, PCAs with high levels bring the opportunity to reduce the AC inductor value to a much smaller value. Since spreading this AC inductor value over the numerous levels is possible in PCA topologies, the individual inductor value drops; in this case study, to below 1 μH. This makes it possible to integrate them directly inside the CSC IPEM. In addition to being integrated directly into the conversion cell itself, the AC inductor becomes so small that it has an outstanding frequency response with a resonant frequency above 120 MHz in our case. This is very efficient because the filtering effect of the components is active over the entire regulation frequency range, making the filter design very reliable.

Of course, more components bring additional costs and reliability issues. However, this work clearly shows that PCA topologies realized with many CSCs offer a significant reduction of AC inductor volume (see Table 3) and of their equivalent DC resistance, and have a very effective impact on the filter transfer function over the entire frequency range of the regulation.

Regarding other important design issues, efficiency is a key point. This has been already investigated in another publication [10] for the case investigated in the present paper of the PCA converter with three CSCs. Please refer to this paper [10] for more information related to efficiency of PCA converters. Cost is also an important issue in power electronics. Many small components are usually more expensive than an equivalent larger one. We have shown here that interleaving control and multilevel topologies bring significant component value reductions, leading to significant cost reductions. It is difficult to say that these positive impacts on cost may mitigate the extra cost required to implement interleaved and multilevel topologies and extra work must be done here to compare both approaches.

At last, if the topology complexity and reliability issues can be mitigated by standardized hardware and firmware, PCA converters could offer very interesting design and manufacturing alternatives to regular converters.

## 5. Conclusions

The paper has presented a comparison between a regular single-phase H-bridge converter cascaded with a DAB converter and a power converter array (PCA) topology, implemented with several conversion standard cells operated under interleaved control. The comparison focused on DM conducted EMI disturbances and highlighted the benefits of the PCA topology compared to a regular H-bridge topology in terms of DM filter needs. This work also showed that the PCA approach enables the AC inductor to be distributed and integrated directly into the CSC comprising the PCA. This considerably increases the performance of the differential mode filter. The results are encouraging. However other important criteria must be compared, such as reliability, efficiency, power density and costs. These points remain to be addressed in future work.

**Author Contributions:** Conceptualization, J.-C.C., Y.L., T.-H.P.; Formal analysis, J.-C.C., T.-H.P., Y.L.; Investigation, T.-H.P., V.-S.N., T.L., A.A. and L.K.; Supervision, J.-C.C. and Y.L.

**Funding:** This research has been supported by CNRS, Grenoble-INP and UGA public institutions.

**Conflicts of Interest:** The authors declare no conflict of interest.

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
