# Peer review of "DC-AC Isolated Power Converter Array. Focus on Differential Mode Conducted EMI"

_electronics, doi:10.3390/electronics8090999_

Round 1

Reviewer 1 Report

Although this paper deals with a rather interesting topic, it has a lot of flaws and points that need significant improvement.

I note some of these below:

In line 27 "(DAB)" is missing. Introduction should be enriched in text and especially in references. The text (suggestions and conclusions) presented in this section do not have the appropriate support by relevant references. In line 41, replace word "than" with "then" What does "ISOP" stand for in the caption of Figure 1; Furthermore, in Figure 1b the red dotted line indicates a DC/AC conversion stage and not a DC/DC. In line 53, replace word "de" with word "the". In lines 85-86:  increasing switching frequency the harmonics order is increased (shifted in higher frequencies). The authors state that in this case the harmonics magnitudes are decreasing which usually is not true. Figures 6 and 7 should have a more informative captions (give better descriptions, note clearly in which signals, the different colors and dot signs are related to). Correct the caption of Figure 8. Add the standard limit line in Figures 9,10, 11 and 12 (where it is missed).

There are many syntax errors scattered throughout the paper and extensive editing of English language is required.

The efficiency, reliability and cost are important aspects of the proposed topology. Since the authors have constructed both the conventional (full bridge) and the PCA converter, they should discuss about these characteristics.

Author Response

Although this paper deals with a rather interesting topic, it has a lot of flaws and points that need significant improvement.

Dear Reviewer, thank you for the review and the comment provided that are helping us improving our work and the corresponding paper. Please find below our actions based on your remarks.
In addition, the entire paper has been checked for grammatical mistakes. Hopefully most of them have been removed.

I note some of these below:

In line 27 "(DAB)" is missing.

DAB has been added in the text.

Introduction should be enriched in text and especially in references.

The introduction and other parts in the paper are more referenced

The text (suggestions and conclusions) presented in this section do not have the appropriate support by relevant references.

We have added several reference to support our statements whenever appropriate.

In line 41, replace word "than" with "then"

Typo corrected

What does "ISOP" stand for in the caption of Figure 1;

This has been defined and explained in the text before the Figure 1. It has also been corrected since it is a IPOS configuration.

Furthermore, in Figure 1b the red dotted line indicates a DC/AC conversion stage and not a DC/DC.

The schematic has been corrected.

In line 53, replace word "de" with word "the".

This has been corrected

In lines 85-86:  increasing switching frequency the harmonics order is increased (shifted in higher frequencies). The authors state that in this case the harmonics magnitudes are decreasing which usually is not true.

Section has been completely modified

Figures 6 and 7 should have a more informative captions (give better descriptions, note clearly in which signals, the different colors and dot signs are related to).

This has been done for the two figures

Correct the caption of Figure 8.

This has been done

Add the standard limit line in Figures 9,10, 11 and 12 (where it is missed).

This has been done. In addition, the theoretical spectrum have been added on the plots.

There are many syntax errors scattered throughout the paper and extensive editing of English language is required.

The paper has been entirely reviewed to improve the English as much as we could

The efficiency, reliability and cost are important aspects of the proposed topology. Since the authors have constructed both the conventional (full bridge) and the PCA converter, they should discuss about these characteristics.

Another paper is dedicated to the efficiency comparison. A reference has been added to that paper that will be published at the coming up EPE conference in September. Together with the reference, a small discussion has been added in order to comment on this and at the same time to keep the length of the paper reasonable. .

Reviewer 2 Report

This paper presents the comparisons of DC-AC isolated full-bridge converter and DC-AC Isolated power converter array. Although the analysis is supported by experimental results, the contribution of the paper is weak as only sensitive analysis has been conducted instead of proposing a general analysis method. Moreover, there are numerous grammatical mistakes. Please find the following comments from the reviewer.

Please provide the full name of the first time use abbreviation, such as the DM, EMI, THD, HF. Page 1, line 41, Please check the grammar "CSCs are than connected..." Page 2, line 48, What is the ISOP? Please provide the full name in the context. Page 2, Section 2, Please present more critical review of the state-of-the-art of DC-AC converters in the literature, especially DC-AC Isolated Power Converter Array topologies related to the work of this paper. Page 2, Table 1, How to obtain the values of the AC inductor value/CSC and the AC inductor with interleaving? Similarly, how to obtain the values in other tables? Any reference? Page 4, line 146, Please check the grammar "In the table 2 are summarized, ..." Page 4, the Figure 3 is not mentioned in the context. Please also provide reference of the photo. The authors are encouraged to carry out more theoretical analysis. 

Author Response

This paper presents the comparisons of DC-AC isolated full-bridge converter and DC-AC Isolated power converter array. Although the analysis is supported by experimental results, the contribution of the paper is weak as only sensitive analysis has been conducted instead of proposing a general analysis method. Moreover, there are numerous grammatical mistakes. Please find the following comments from the reviewer.

Dear Reviewer, thank you for the review and the comment provided that are helping us improving our work and the corresponding paper. Please find below our actions based on your remarks.
In addition, the entire paper has been checked for grammatical mistakes. Hopefully most of them have been removed.

Please provide the full name of the first time use abbreviation, such as the DM, EMI, THD, HF.

This has been corrected throughout the whole paper exception made in the abstract to keep it short enough.

Page 1, line 41, Please check the grammar "CSCs are than connected..."

Typo corrected

Page 2, line 48, What is the ISOP? Please provide the full name in the context.

Corrected in the texte and in the figure caption. IPOS is now defined and explained in the text before figure 1.  

Page 2, Section 2, Please present more critical review of the state-of-the-art of DC-AC converters in the literature, especially DC-AC Isolated Power Converter Array topologies related to the work of this paper.

Several references have been added

Page 2, Table 1, How to obtain the values of the AC inductor value/CSC and the AC inductor with interleaving? Similarly, how to obtain the values in other tables? Any reference?

Equations have been added to explain how the inductor values are obtained in table 1. All data listed in Table 2 are coming from practical characterization. This has been mentioned in the table Caption.

Page 4, line 146, Please check the grammar "In the table 2 are summarized, ..."

The sentence has been rephrased.

Page 4, the Figure 3 is not mentioned in the context. Please also provide reference of the photo.

Mention to figure 3 has been added as well as reference to the component. The picture is original, not coming from another paper.

The authors are encouraged to carry out more theoretical analysis.

In figures 9-12, the experimental spectrum are all compared with the theoretical one coming from the modeling technique recalled and used in this paper. The frequency domain modeling technique itself has not been developed further because it is based on previous publications. A reference has been added.

Inductor calculation equation for the tables have been added in the paper in order to explain the way values are obtained.

Reviewer 3 Report

The authors have focused on DC AC isolated full bridge converter and a PCA topology. The paper is very well written and i have only few suggestions. 

The references are not adequate. This can be improved. For eg. "DC to AC isolated step up bidirectional converters.... many applications such as electric mobility, remote applications, but also 22 renewable energy storage." Sentences like these need references for applications of mobility, remote applications and energy storage. This needs to be repeated throughout the paper.   The curves in Figs 8-13, should be overlapped with a smoother curve, in addition to the raw data, which can be obtained by applying a smoothing function.  If the authors can add a plot comparing models and experiments, it would be useful to the readers. 

If the authors work on these suggestions, I recommend publication. 

Author Response

The authors have focused on DC AC isolated full bridge converter and a PCA topology. The paper is very well written and i have only few suggestions.

Dear Reviewer, thank you for your positive feedback and the remarks that are useful to improve our paper. Below are a few comments on your suggestions.

The references are not adequate. This can be improved. For eg. "DC to AC isolated step up bidirectional converters.... many applications such as electric mobility, remote applications, but also 22 renewable energy storage." Sentences like these need references for applications of mobility, remote applications and energy storage. This needs to be repeated throughout the paper.

We have added several references to illustrate the statements when necessary.

The curves in Figs 8-13, should be overlapped with a smoother curve, in addition to the raw data, which can be obtained by applying a smoothing function.  If the authors can add a plot comparing models and experiments, it would be useful to the readers.

The theoretical plots have been overlapped with the experimental results.

Round 2

Reviewer 1 Report

The work has been greatly improved both in terms of scientific soundness and in terms of grammar and spelling errors which were spread out in the previous version of the paper.

The authors have replied satisfactorily to my comments, adding a lot of relevant references, clarifying the presented simulation and experimental results and correcting typographical, syntax and spelling errors.

Although English is not my native language, I still believe that there are several syntax and spelling errors which should be corrected before publication.

Reviewer 2 Report

Thanks for considering the comments from the reviewer. The paper has been improved a lot compared to the first submission. Now it can be published.